# Effect of Nutritional Deprivation after Sleeve Gastrectomy on Bone Mass, Periostin, Sclerostin and Semaphorin 4D: A Two-Year Longitudinal Study

**DOI:** 10.3390/nu15204310

**Published:** 2023-10-10

**Authors:** Laurent Maïmoun, Safa Aouinti, Marion Puech, Patrick Lefebvre, Mélanie Deloze, Pascal de Santa Barbara, Jean-Paul Cristol, Séverine Brabant, Thomas Gautier, Marius Nedelcu, Eric Renard, Marie-Christine Picot, Denis Mariano-Goulart, David Nocca

**Affiliations:** 1Département de Biophysique, Université Montpellier, Service de Médecine Nucléaire, Hôpital Lapeyronie, 371, Avenue du Doyen Gaston Giraud, CHU de Montpellier, CEDEX 5, 34295 Montpellier, France; d-mariano_goulart@chu-montpellier.fr; 2Physiology and Experimental Medecine of the Heart and Muscles (PhyMedExp), Université de Montpellier, INSERM, CNRS, 34295 Montpellier, France; pascal.de-santa-barbara@inserm.fr; 3Unité de Recherche Clinique et Epidémiologie, CHU de Montpellier, Université de Montpellier, 34295 Montpellier, France; s-aouinti@chu-montpellier.fr (S.A.); mc-picot@chu-montpellier.fr (M.-C.P.); 4Service de Chirurgie Digestive A, Hôpital Saint Eloi, CHU de Montpellier, 34295 Montpellier, France; m-puech@chu-montpellier.fr (M.P.); m-deloze@chu-montpellier.fr (M.D.); d-nocca@chu-montpellier.fr (D.N.); 5Department of Endocrinology and Diabetes, Lapeyronie Hospital, CHU de Montpellier, University of Montpellier, INSERM, CNRS, 34295 Montpellier, France; p-lefebvre@chu-montellier.fr (P.L.); e-renard@chu-montpellier.fr (E.R.); 6Laboratoire de Biochimie, Hôpital Lapeyronie, CHU de Montpellier, 34295 Montpellier, France; jp-cristol@chu-montpellier.fr; 7Laboratoire des Explorations Fonctionnelles, Hôpital Necker Enfants Malades, APHP, 75015 Paris, France; severine.brabant@aphp.fr; 8Clinique Saint Jean, 34430 Saint-Jean-de-Vedas, France; t.gauthier@gmail.com; 9Clinique Saint-Michel, 83100 Toulon, France; nedelcu.marius@gmail.com

**Keywords:** areal bone mineral density, markers of bone turnover, sleeve gastrectomy, periostin, sclerostin, semaphorin 4D, bone loss

## Abstract

Bariatric surgery induces bone loss, but the exact mechanisms by which this process occurs are not fully known. The aims of this 2-year longitudinal study were to (i) investigate the changes in areal bone mineral density (aBMD) and bone turnover markers following sleeve gastrectomy (SG) and (ii) determine the parameters associated with the aBMD variations. Bone turnover markers, sclerostin, periostin and semaphorin 4D were assessed before and 1, 12 and 24 months after SG, and aBMD was determined by DXA at baseline and after 12 and 24 months in 83 patients with obesity. Bone turnover increased from 1 month, peaked at 12 months and remained elevated at 24 months. Periostin and sclerostin presented only modest increases at 1 month, whereas semaphorin 4D showed increases only at 12 and 24 months. A significant aBMD decrease was observed only at total hip regions at 12 and 24 months. This demineralisation was mainly related to body weight loss. In summary, reduced aBMD was observed after SG in the hip region (mechanical-loading bone sites) due to an increase in bone turnover in favour of bone resorption. Periostin, sclerostin and semaphorin 4D levels varied after SG, showing different time lags, but contrary to weight loss, these biological parameters did not seem to be directly implicated in the skeletal deterioration.

## 1. Introduction

It has been well-established that patients with obesity present higher areal bone mineral density (aBMD) than normal-weight subjects [1,2,3,4,5]. In a large cross-sectional study that included more than 500 obese patients with ages ranging from 18 to 82 years, we observed that this phenomenon was accentuated with obesity severity and age and in women compared to men [6]. The increase in body weight that is associated with bone mass gain [6,7] may accentuate the mechanical loading on the skeleton, and it has been identified as the major factor influencing the gain of bone mass in this population [1]. Indeed, bone is an adaptive tissue that has the capacity to modify its microarchitecture and mass in response to a mechanical stimulus [8]. Nevertheless, the concomitant increase in aBMD in non-weight-bearing bone, such as the radius [3,6,7], suggests that the circulating molecules related to obesity, in addition to the gravitational forces associated with increased body weight, also act on the bone tissue in these patients. Among these molecules produced by adipose tissue, increased leptin production [9,10] and oestrogen synthesis by adipocytes [11,12] have been found to have positive actions on bone mass. Muscle-secreted factors, defined as “myokines”, also biochemically affect bone metabolism in both paracrine and endocrine manners [13]. In the obese population, we recently demonstrated that this action may be mediated mainly by irisin and not by follistatin or myostatin [7]. Given these findings, a lesser favourable environment for bone mass may result from a rapid and intense loss of body weight after bariatric surgery (BS) in obese patients. Numerous studies, mostly based on Roux-en-Y gastric bypass (RYGB), the most frequently performed bariatric procedure worldwide until very recently [14], have reported that this surgery has a negative impact on bone health that is characterised by a decrease in aBMD, an increase in bone turnover markers and an alteration in bone microarchitecture parameters [15,16,17]. The resulting increase in the risk of fractures [18] should prompt us to better identify the underlying mechanisms involved. In fact, the negative effects of BS on bone metabolism are multifactorial and may involve a deficiency in nutritional factors, alterations in gut-derived hormones and adipokines and changes in body composition [19]. Moreover, the strong association between the amount of weight loss and the extent of postoperative bone loss also suggests an important role of the reduced mechanical loading in bone health [20]. This process may be mediated by the increase in sclerostin levels [21]. In addition, periostin, a matricellular protein of 90 kD secreted by osteocytes and osteoblasts [22,23], which plays a crucial role in bone formation partly by promoting osteoblast differentiation and proliferation [24], may be also modified. Experimental preclinical studies have demonstrated that an increase in mechanical constraints using, for example, an axial compression load on mouse tibia [23] or intensive training [22] results in the overexpression of periostin and the downregulation of sclerostin [23]. Until now, lower periostin values were observed in young obese women [25] and subjects with higher BMIs [26]. Higher aBMD, which reduces the perception of external mechanical strain [25] and the systemic inflammation observed in obese patients, may explain the reduction in periostin levels [27,28]. Other biological parameters, such as semaphorin 4D (SEMA4D), may potentially affect bone loss after BS, given that serum levels in osteoporotic postmenopausal women were found to be negatively correlated with lumbar spine aBMD and bone turnover markers [29]. SEMA4D is a transmembrane protein secreted by osteoclasts that binds to its receptor PlexinB1 on osteoblasts to inhibit their differentiation and function [30]. However, until now, the implication of periostin and SEMA4D in bone loss following BS has not been investigated.

The aims of this study were to (i) follow the variation in aBMD in various weight-bearing and non-weight-bearing bone sites after sleeve gastrectomy (SG) over a 24-month period, (ii) determine the variation in periostin, sclerostin and SEMA4D levels in acute weight loss and weight stabilisation periods and (iii) determine the main clinical and biochemical parameters associated with aBMD variation.

## 2. Materials and Methods

### 2.1. Subjects and Method

This study followed a longitudinal design. The evaluation was performed the day before the operation (baseline) and 1, 12 and 24 months after the procedure. Study approval was obtained from the Regional Research Ethics Committee (Sud Méditerranée I, Marseille, France; ID RCB: 2015-A01047-42) and permission for the clinical trials was granted by the French Health Products Safety Agency. Written informed consent was obtained from all participants. The clinical trial number is NCT02712086.

#### 2.1.1. Subjects

Eighty-three Caucasian patients were recruited from a waiting list of candidates for obesity surgery at the Obesity Reference Centre, CHU Montpellier, France, from November 2016 to June 2018. The data were collected from November 2016 to June 2020. Patients were selected for surgery if other treatments for weight loss had failed and if BMI was >40 kg/m^2^ (severe obesity) or ≥35 kg/m^2^ with the presence of obesity-related comorbidities such as type 2 diabetes (T2D), arterial hypertension or sleep apnoea syndrome. These patients had a longstanding history of obesity (more than 4 years) and none of the included patients had undergone BS. Exclusion criteria were pregnancy, acute medical treatment and physical handicap (amputation, neurological lesion, orthopaedic prosthesis) that might interfere with bone mass. None of the patients were taking a medication known to affect bone metabolism or presented with primary amenorrhoea. Physical activity levels were not specifically determined, but none were participating in a training program on the day of inclusion. Moreover, patients with a body weight > 190 kg or height > 192.5 cm were also excluded due to limitations of the densitometry device. Medical history and current medications were obtained by questionnaire. All the bariatric surgery procedures were SG, which consists of resecting most of the greater curvature to reduce gastric size and leave a narrow stomach tube. SGs were performed in a single institution.

#### 2.1.2. Methods

For each visit, weight and standing height were measured and BMI was calculated as weight (kg) divided by the square of height (m).

#### 2.1.3. Comorbidities Were Defined According to the Usual Definitions

T2D was defined as HbA1c ≥ 6.5% and/or fasting glycaemia ≥7 mmol/L and/or antidiabetic treatment [31].

Arterial hypertension (HTA) was defined as systolic blood pressure >140 mmHg and/or diastolic blood pressure >90 mmHg and/or use of antihypertensive medications [32].

Vitamin D deficiency was defined as a 25-OH vitamin D level of <20 ng/mL, and vitamin D insufficiency was defined as a 25-OH vitamin D level of 21–29 ng/mL [33].

#### 2.1.4. Assays

Blood samples (25 mL) were collected in fasting conditions in the morning (8:30–9:00 a.m.) in sterile chilled tubes by standard venepuncture technique. The samples were allowed to clot at room temperature and were then centrifuged at 2500× *g* rpm for 10 min at 4 °C. Serum samples were stored at −80 °C until analysis. All samples were run in duplicate and analysed in a single session to reduce inter-assay variation. The dates of the female patients’ last menses were not recorded, and hormonal values were thus obtained at an unsynchronised menstrual stage.

Calcium, phosphorus, albumin, 25-OH vitamin D and intact parathyroid hormone (iPTH) were routinely analysed by an automated Cobas (Roche Diagnostic, Mannheim, Germany). For the markers of bone remodelling, serum samples were assayed using an IDS-iSYS system (Immunodiagnostic Systems, Boldon, UK) analyser for osteocalcin (OC) and type I-C telopeptide breakdown products (CTX). The inter- and intra-assay coefficients of variation (CVs) for the latter two parameters were <7%.

Serum sclerostin, periostin and SEMA4D were measured with quantitative sandwich ELISA kits from Biomedica (Vienna, Austria; references: BI-20492, BI-20433 and BI-20405). The intra-assay and inter-assay CVs were ≤10%, ≤6% and ≤11%, respectively.

#### 2.1.5. Areal Bone Mineral Density

DXA (Hologic Horizon A, Hologic, Inc., Waltham, MA, USA) measured aBMD (g/cm^2^) of the whole body and at specific bone sites: the anteroposterior lumbar spine (L1–L4), the dominant arm distal radius and total hip. All scanning and analyses were performed by the same operator to ensure consistency after following standard quality control procedures. Quality control for DXA was checked daily by scanning a lumbar spine phantom consisting of calcium hydroxyapatite embedded in a cube of thermoplastic resin (Hologic DXA quality control phantom). Identical and accurate positioning of the region of interest was ensured by superimposing the image from the first session (before SG) on the image from the second, third and fourth sessions (1 month, 1 year and 2 years after SG). The CVs were 1.25% for spine, 1.6% at the total hip, 1.94% for femoral neck [34]. For whole body, total hip, lumbar spine and one-third radius, T-scores and Z-scores were obtained from reference data. The T-score describes the number of standard deviations (SDs) by which BMD differs from the mean value expected in young healthy individuals, while the Z score describes the number of SDs by which BMD differs from the mean value expected for age and sex.

##### Osteopenia and Osteoporosis Definitions

Osteopenia and osteoporosis were defined according to WHO criteria [35]: osteopenia by the lowest T-score at the spine or hip between −1 and −2.5 SD and osteoporosis by a T-score at the spine or hip ≤−2.5 SD. Normal bone density was defined by a T-score at the spine and total hip ≥−1 SD.

#### 2.1.6. Statistical Analysis

The characteristics of the individuals were described with numbers and proportions for categorical variables and with means and SDs for quantitative variables after normality testing with the Shapiro–Wilk test.

Paired Wilcoxon or paired Student’s tests were used, depending on the normality of the distribution, to compare the relative variations (100 × (measure 2 − measure 1)/measure 1) between baseline and 1, 12 and 24 months for the patients’ biological parameters and between baseline and 12 and 24 months for aBMD.

Linear regression models were performed to analyse the influence of the percentage (%) of the relative variations in various anthropometrical and biological variables on whole body aBMD, total hip aBMD, lumbar spine aBMD and radius aBMD between baseline and 12 and 24 months. Backward selection based on the Akaike Information Criterion was applied in all the multivariable models. Collinearity between factors was tested with variance inflation factors. To test the validity of the model, the normality of residues was tested with the Shapiro–Wilk test.

All analyses were two-tailed, with a *p*-value of <0.05 considered statistically significant. SAS^®^ Enterprise Guide software (version 8.2, SAS Institute, Cary, NC, USA) was used to perform the analyses, and graphs were generated using R statistical software (www.r-project.org, accessed on 7 October 2023, version 4.1.3) with the ggplot2 package (version 3.4.0).

## 3. Results

### 3.1. Anthropometric Parameters

At baseline, this study included 83 patients with obesity, 21 men (25.3%) and 62 women (74.7%); of the women, 16 (25.8%) were menopausal (mean age of menopause 47.7 ± 5.2 years). The mean age was 40.8 ± 12.3 years (ranging from 18.4 to 60.0 years), and the mean baseline BMI was 40.7 ± 4.2 kg/m^2^ (Table 1). The mean weight loss was −10.0 ± 2.6 kg after 1 month, −32.5 ± 9.6 kg after 12 months and −31.1 ± 11.5 kg after 24 months (all *p* < 0.001). During the follow-up, some patients withdrew from this study (Figure 1): at 1 month, two patients (one use of protein supplementation before the surgery and one withdrawal of informed consent); at 12 months, seven patients (one patient died, one became pregnant, and five were lost to follow-up or no longer wished to participate in this study); and at 24 months, 16 patients (patients lost to follow-up or no longer wishing to participate in this study).

### 3.2. Biological Parameters

All the biological parameter variations are presented in Table 1 and Figure 2. No initial calcium or phosphorus variation occurred, whereas calcium was modestly decreased at 12 and 24 months (−0.9 and −1.4%, respectively, *p* < 0.001 for both) and phosphorus transiently increased at 12 months (11.3%, *p* < 0.001). Serum iPTH and albumin levels remained relatively stable throughout the 24-month period, whereas 25(OH)D3 levels were significantly increased at 12 and 24 months (*p* < 0.001) after surgery. The prevalence of patients presenting 25(OH)D3 deficiency was 39.8% before surgery and 37.5% at 1 month, 14.9% at 12 months and 18.3% at 24 months after surgery.

Osteocalcin and especially CTX values increased after 1 month and peaked at 12 months. From 12 to 24 months, these two bone turnover markers decreased, but the values did not return to baseline levels. Although these two parameters presented similar kinetic profiles, the magnitude of difference compared to baseline values was higher for CTX than for osteocalcin. At 1 month, periostin and sclerotin values showed a modest but significant 12% increase (*p* < 0.001 for both) but had returned to basal values after 12 months. Sclerostin levels continued to decrease after 12 months to reach a value lower than baseline at 24 months (*p* < 0.010). No initial variation in SEMA4D levels was observed at 1 month, although a modest but significant decrease (10.5% and 11.8%, respectively) occurred after 12 and 24 months (*p* < 0.001 for both).

### 3.3. Areal Bone Mineral Density

The baseline and changes in aBMD values at 12 and 24 months after SG are presented in Table 2 and Figure 3. At baseline and for all bone sites, patients with obesity presented Z-score values higher than zero, indicating that the aBMD was above the age- and gender-related reference values. This bone adaptation was particularly marked at total hip (1.09 ± 0.97 SD) and at one-third radius (2.11 ± 1.45 SD).

After 12 and 24 months, a relative stability of aBMD was observed at whole body, lumbar spine and one-third radius, whereas a significant decrease was observed at femoral neck (−5.4% and −5.8%) and total hip (−6.6% and −7.9%). Nevertheless, at total hip, the % relative variation ranged from −19.5 to 2.9% at 12 months and −19.4 to 2.9% at 24 months, both denoting large inter-individual variations. After 12 and 24 months, mean Z-scores remained above zero for all bone sites. Using the WHO classification, 2.4% and 10.8% of patients were diagnosed with osteopenia at baseline when total hip or lumbar spine were used for the diagnosis, whereas, respectively, none and one was diagnosed with osteoporosis. At lumbar spine, the prevalence of osteopenia or osteoporosis did not vary postoperatively, but at total hip, the prevalence of osteopenia increased modestly by 6.6% and 8.3% at 12 and 24 months, respectively.

### 3.4. Predictors of aBMD Variation

Univariate analysis revealed that various parameters (basal or variations), depending on the bone site (i.e., whole body, hip regions, lumbar spine or radius) or the time point (i.e., 12 or 24 months) were associated with % relative variation aBMD. These included iPTH, osteocalcin, periostin, sclerostin, SEMA4D, 25(OH)D3, gender and weight. However, when multivariate analysis was performed, only weight change appeared strongly and independently associated with % relative variation aBMD at weight-bearing bone (i.e., hip regions), as well as non-weight-bearing bone, including lumbar spine (at 12 months only) and radius (12 and 24 months). To a lesser extent, other parameters such as iPTH, osteocalcin, periostin, sclerostin, SEMA4D, albumin, 25(OH)D3 and CTX also appeared partially associated with weight change, depending on the bone site or the time lag of investigation. The influence of various anthropometric and biological variables on aBMD variation at 12 and 24 months are presented in detail in Appendix A.

## 4. Discussion

This study was designed to improve our knowledge on the factors that may influence aBMD loss after SG, and it especially sought to determine whether the reduction in mechanical loading on the skeleton due to the extreme body-weight loss would influence periostin, sclerostin and SEMA4D levels. The main findings showed a persistent decrease in aBMD throughout the 24 months after SG, but with a site-dependent effect. The bone loss was concomitant to an uncoupling between bone formation and bone resorption activities that persisted despite the attainment of a new steady state in weight. Periostin, sclerostin and SEMA4D levels were also modified after SG, but with different time lags.

### 4.1. aBMD

As suggested by Matos et al. [36], BS provides a unique condition in which an extreme variance from high to reduced body-weight-bearing is perceived by the individual in a very short period. Consequently, biomechanical effects should occur mainly in bone areas submitted to higher load bearing. In accordance with this assumption, at 12 months, a noticeable and significant decrease in aBMD was mainly observed at femoral neck (−5.4%) and total hip aBMD (−6.6%), while at whole body, lumbar spine and radius, the aBMD variations were within the precision error of the DXA measurement. Between 12 and 24 months, the loss continued at femoral neck and total hip, but the decrease was slowed down compared to the first 12 months (−5.8% and −7.9%, respectively). Using a similar surgical procedure, Hofso et al. [16] reported a similar aBMD loss around 5% at femoral neck and 7.8% at total hip, while at lumbar spine and whole body, minor variations occurred after 12 months. A recently published meta-analysis including 22 studies [37] noted an obvious aBMD loss at femoral neck and none at lumbar spine. This conclusion should be considered with caution, as some of the included studies reported an aBMD decrease at lumbar spine [21]. The type of population under study or the degree of weight loss may partially explain these divergent results. Moreover, a recent meta-analysis underlined that, having QCT and HR-pQCT as a reference, DXA was found to significantly underestimate lumbar spine post-RYGB aBMD losses [38].

To our knowledge, although the analysis of non-weight-bearing bone (i.e., radius) may improve our understanding of the potential effects of endocrine modifications independently of mechanical loading, no data on aBMD variations following SG are available. After 12 and 24 months, we observed a limited decrease in aBMD, around 2% at radius, but the Z-score remained much higher compared to the reference values (+1.95 SD). Using another technique (i.e., HR-pQCT) and surgical procedure (i.e., RYGB), Shanbhogue et al. [15] also reported a minimal variation in volumetric BMD (vBMD) at the trabecular and cortical compartments of the radius after 12 months, while at 24 months, a decline in total vBMD occurred, mainly due to an alteration in the trabecular compartment [15]. This suggests that an energy deficit and changes in adipose or gastrointestinal hormones could be additional contributors to BS-induced bone loss [39], but probably to a smaller extent than the reduction in mechanical loading.

In fact, although a noticeable aBMD decrease occurred after SG at the hip region, our data do not conclusively support an increased prevalence of osteopenia or osteoporosis over the 24 months. Several studies also reported a very low incidence of T-scores ≤ −2.5 SD according to the standard DXA definition [40,41]. In our study, it is further interesting to note that the bone loss, despite being low, continued after 12 months, whereas body weight was already stabilised. This finding supports the hypothesis that aBMD loss and fracture risk are not dependent only on weight loss and rather increase with observation time [15,42].

### 4.2. Bone Markers

The use of biochemical markers of bone turnover may have helped to elucidate the mechanisms involved in this bone loss. After SG, the variations in bone formation and bone resorption seem to have the same kinetic profile characterised by an initial increase from the third month, a peak at around 12 months and a decrease thereafter, but with values at 24 months always remaining higher than the basal values. Nevertheless, the bone formation marker (i.e., osteocalcin) level increased to a lesser extent than the bone resorption marker (i.e., CTX), suggesting an uncoupling of bone cell activities in favour of bone resorption. This process may explain the reduction in aBMD in the first 24 months. Moreover, the persistence of a high degree of bone turnover at the end of the follow-up may also suggest that aBMD loss is likely to continue for a longer time after BS and that aBMD monitoring by DXA should thus be performed beyond 24 months in these patients. Our results fully confirmed the recently published results of Paccou et al. [17], who studied a small group of 32 patients of both sexes and women with different menopausal status and found a similar profile of variation in bone turnover using procollagen type I *N*-terminal propeptide (PINP), another bone formation marker, and CTX after 12 and 24 months. Muschitz et al. [21] performed a longitudinal study of 52 premenopausal women, using multiple evaluation points, and also showed an ongoing increase in CTX and PINP from 1 month to 12 months, and these values remained elevated at 24 months. Lastly, Hofso et al. [16] studied obese patients of both sexes with T2D and women with different menopausal status and showed a progressive increase in PINP and CTX from 5 to 52 weeks after SG. Up to now, it has not been clear whether the type of surgery has specific effects on bone marker levels, with some studies reporting a greater detrimental effect of RYGB compared to SG [16,17] and others finding no difference [21].

Numerous factors may influence bone cell activities, including mechanical and endocrine factors. It was suggested that the increase in bone turnover following BS may be related to the increase in calcium and parathyroid hormone [43]. However, as also previously observed, we reported no variation in either of these biological parameters [15,21,39], even though pre-surgical and post-surgical elevated PTH was also reported [21,43]. At baseline, the obese patients presented lower mean values of vitamin D (<30 ng/mL) and, although the values increased with duration after surgery, they remained at the lower limits [33]. As observed in our study, hypovitaminosis D is frequently reported in obese patients, possibly partially due to sequestration of vitamin D in subcutaneous and visceral fat [44]. It has been hypothesized that the reduction in adipose tissue, which releases stored vitamin D into the circulation associated with high vitamin D supplementation [15,39], may partially explain the increase in vitamin D but strongly suggests the need for adapted vitamin D supplementation. It was reported that 39% of patients presented postoperative vitamin D deficiency despite daily multivitamin supplementation after a restrictive procedure such as SG [45].

Conversely to calciotropic hormones and the routinely used bone turnover markers, periostin, sclerostin and SEMA4D have been poorly or never evaluated after SG [17,21,39]. We reported for the first time an initial, transient and proportional increase in periostin and sclerostin at 1 month. However, over the first 12 months, and although OC and CTX remained elevated, periostin and sclerostin returned to basal values, suggesting their limited effect on bone cell differentiation and activities. A similar conclusion could be drawn after 24 months, where a modest decrease was observed for sclerostin alone. Our results concerning sclerostin fully confirm those of Paccou et al. [17] at 12 and 24 months, even though unfortunately no investigations in the acute phase were available in their study. In premenopausal women, Muschitz et al. [21] reported that sclerostin increased from 1 month after BS, peaked at 6 months and remained higher than baseline up to 24 months, although the value decreased continuously. An increase in sclerostin was also observed in adolescents 12 months after SG [39]. The inconsistent results for sclerostin at 12 and 24 months following BS may be related to patient characteristics or the intensity of body-weight loss, but they seem to be independent of the type of surgery [17,21]. The acute reduction in mechanical loading induced by a 10 kg weight loss [21] associated with the limited daily physical activity [46] during the first month following BS may explain the increase in sclerostin levels. This assumption is based on experimental studies that demonstrated that sclerostin expression is upregulated under conditions of mechanical unloading and downregulated by mechanical stimulation [47]. However, the involvement of sclerostin in post-surgical bone loss after BS remains questionable because its physiological action is to suppress osteoblast activity [48], which has not been currently or previously observed [16,17,21].

In parallel, an unexpected transient increase in periostin levels at 1 month was also observed. In fact, experimental studies demonstrated that a reduction in mechanical loading is associated with a concomitant deterioration in bone structure and a decrease in periostin gene expression [49]. Conversely, an increase in mechanical constraints using an axial compression load on mouse tibia induced the overexpression of periostin and the downregulation of sclerostin [23]. In humans, the effect of mechanical loading or unloading on circulating periostin levels remains more controversial [50,51]. However, in the same unexpected way as after SG, we also reported a higher level of periostin in persons with spinal cord injury during the acute phase of immobilisation, when the bone loss is maximal, compared to the value in the chronic phase, when the bone loss is reduced [50]. We [50] and others [52] interpreted this high periostin level as a “protective mechanism” or as a “compensatory mechanism” in the case of low bone mass or during an acute bone loss phase. The transient increase in periostin levels at 1 month—a period when weight loss and thus the variation in mechanical loading are maximal—could go in this direction. Nevertheless, the normalisation of sclerostin and periostin levels with the post-surgery duration, whereas bone turnover remains intense, argues for a limiting effect of periostin/sclerostin on induced SG-related bone loss.

From a more exploratory perspective, we report the SEMA4D variations after SG for the first time and note that the kinetics were characterised by a weak but significant decrease only at 12 and 24 months. SEMA4D is produced by osteoclasts and suppresses the differentiation and activity of osteoblasts [30,53], which does not coincide with the higher post-surgery bone formation values compared with baseline. We must, however, bear in mind that the role of this transmembrane protein in human bone has been little investigated. In one study, a higher serum SEMA4D level was observed in postmenopausal osteoporosis patients compared to healthy controls, and the concentrations were negatively correlated with lumbar spine aBMD [29], bone alkaline phosphatase and bone Gla-protein and positively correlated with markers of bone resorption (TRACP-5b and NTX) [29]. Similarly, Anastasilakis et al. [54] reported a trend in postmenopausal women towards a negative correlation between SEMA4D levels and aBMD measured at lumbar spine but not at femoral neck.

### 4.3. Factors Influencing aBMD Loss

Although the bone loss appeared limited in terms of localisation (i.e., hip region) and intensity (~8%), a noticeable inter-individual variability was observed, which might be of help in identifying the factors influencing aBMD after SG. This may also be of clinical interest because fracture risk seems to increase in patients after BS [18]. Multivariate analysis clearly showed that weight change appeared to be the main parameter influencing aBMD. Interestingly, this effect was observed at weight-bearing (i.e., hip) and non-weight-bearing (lumbar spine and radius) bone sites, suggesting a systemic effect of weight loss that does not appear to be mediated by sclerostin, periostin or SEMA4D. Age, gender and most of the biological parameters (iPTH and osteocalcin, except for a few sites and time points) were also not associated with the aBMD variations. Previous studies have also found that weight change correlated with femoral neck, total hip, lumbar spine and whole body aBMD loss [21,55,56,57]. However, other studies have observed that the effect of weight loss on aBMD loss was not independent of the effect of type of surgery [16]. These findings indicate that gastric bypass per se elicits bone turnover and induces bone loss, and they do not support the idea that reduced aBMD after gastric bypass is solely a physiological adaption to a lower body weight [16].

Taking into account the potential deleterious effects of weight loss, which reduces the mechanical loading applied on the skeleton, it will be interesting to evaluate strategies for restoring bone loading. In addition, previous clinical trials have demonstrated that physical exercise, combined or not with adequate supplementation (vitamin D, calcium and protein), is a practical approach to reduce the bone loss caused by bariatric surgery [58,59]. Other studies are necessary to demonstrate whether this approach should be generalized for all patients undergoing bariatric surgery. The optimal components of physical exercise (frequency, intensity, type and duration) to improve these effects on bone health will also need to be determined, knowing that a low-dose exercise program had no favourable impact [60].

### 4.4. Limitations and Strengths

We are aware that despite the lengthy follow-up of this study, an age-dependent aBMD loss may have occurred, but in the absence of a control group, we cannot be sure. However, the lack of aBMD variation at the lumbar spine and the reduction in bone turnover with time may suggest a limited effect of age. Our observational results are only generalizable to the same type of surgery—i.e., SG—as variations in aBMD between different surgical approaches, in particular RYGB, have been demonstrated [16,37]. Moreover, despite the relatively long time covered by our longitudinal study, we cannot exclude the possibility that other skeletal changes occur over a longer period, and this should be considered in future studies. Conversely, this study presents several strengths: (i) a long-time follow-up from the acute body weight loss period until weight stabilisation, (ii) a population that is representative of the current French population of obese patients who have undergone SG [61], (iii) a population limited only to patients with SG, which is currently the most frequently performed BS technique [38] and (iv) the concomitant evaluation of aBMD, bone turnover markers and biological markers that have been little or not yet analysed, thus providing an overview of the impact of SG on the skeleton in obese patients.

## 5. Conclusions

Our results demonstrated that SG induces an acute and sustainable high bone turnover during the first 24 months after surgery. This imbalance between bone formation and resorption activities is likely at the origin of the aBMD loss predominantly observed at mechanical-loading bone sites (i.e., hip and femoral neck). Periostin, sclerostin and SEMA4D levels varied after SG with different time lags, but contrary to weight loss, these biological parameters did not seem implicated in the skeletal deterioration.

## Figures and Tables

**Figure 1 nutrients-15-04310-f001:**
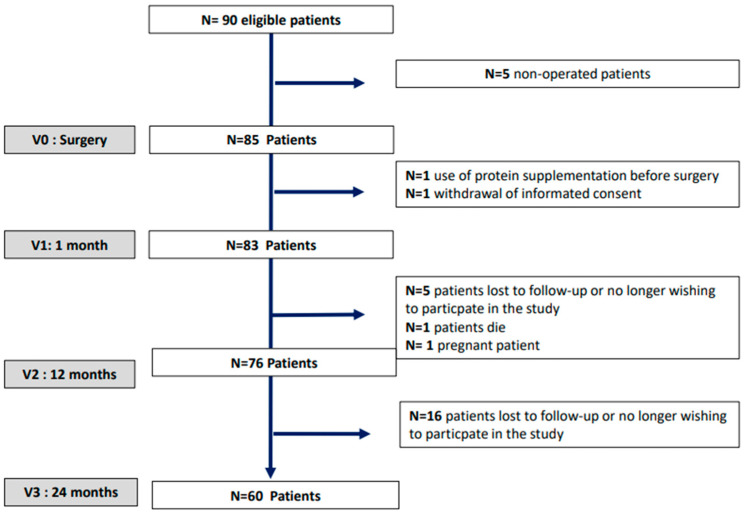
Flowchart of the population studied.

**Figure 2 nutrients-15-04310-f002:**
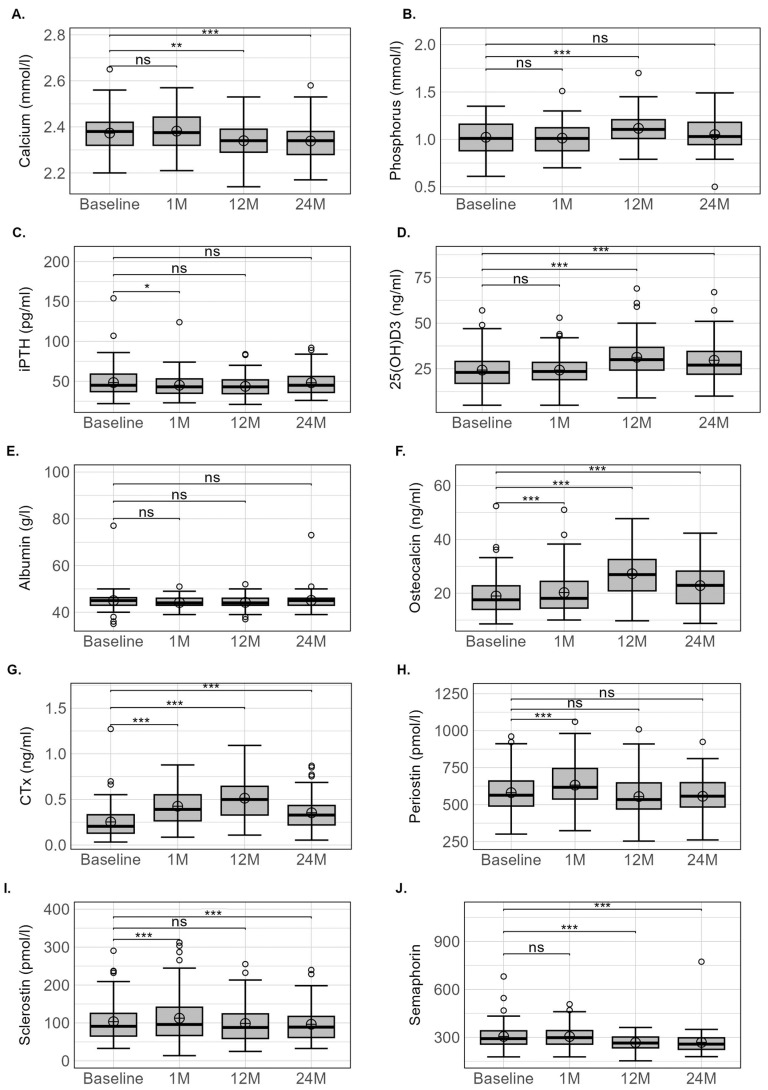
Variation in biological parameters: calcium (**A**), phosphorus (**B**), iPTH (**C**), 25(OH)D3 (**D**), Albumin (**E**), osteocalcin (**F**), CTx (**G**), Periostin (**H**), sclerostin (**I**) and semaphoring (**J**) at different time points (baseline, 1, 12 and 24 months) following sleeve gastrectomy. iPTH: intact parathyroid hormone; 25(OH)2D3: 25 vitamin D; CTX: type I-C telopeptide breakdown products. Lower whisker represents the smallest observation ≥ lower hinge − 1.5 × IQR. Upper whisker represents the largest observation ≤ upper hinge + 1.5 × IQR. Lower and upper hinges represent the 25 and the 75% quartile values, respectively. Open circles represent the mean value at each time point. Data beyond the end of the whiskers are called outlier points and are plotted individually. IQR: interquartile range; * indicates a significant variation for *p* < 0.05, ** for *p* < 0.01, and *** for *p* < 0.001.

**Figure 3 nutrients-15-04310-f003:**
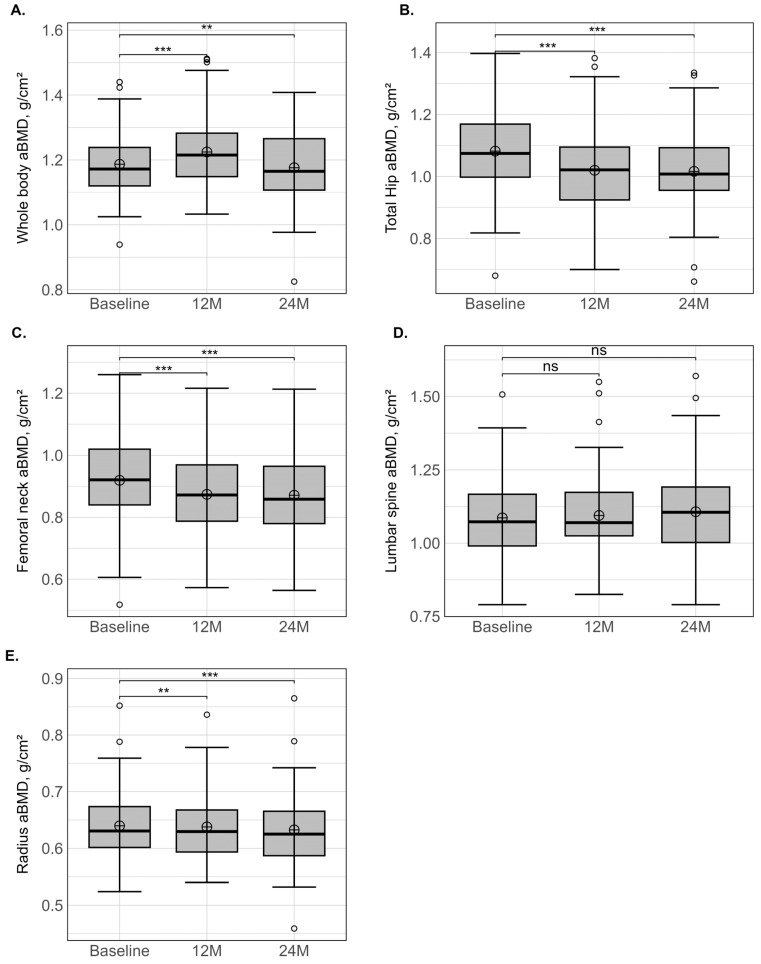
Variation in areal bone mineral density (g/cm^2^) in whole body (**A**), total hip (**B**), femoral neck (**C**), lumbar spine (**D**) and radius (**E**) at different times (baseline, 12 and 24 months) measured at whole body and various localised bone sites. Lower whisker represents the smallest observation ≥ lower hinge − 1.5 × IQR. Upper whisker represents the largest observation ≤ upper hinge + 1.5 × IQR. Lower and upper hinges represent, respectively, the 25 and 75% quartile values. Open circles represent the mean value at each time point. Data beyond the end of the whiskers are called outlier points and are plotted individually. IQR: interquartile range; ** indicate a significant variation for *p* < 0.01, and *** for *p* < 0.001.

**Table 1 nutrients-15-04310-t001:** Biological parameters at baseline, 1, 12 and 24 months after sleeve gastrectomy.

	Baseline (n = 83)	1-Month (n = 83)	12-Months (n = 76)	24-Months (n = 60)	Relative % Variation (Δ 1m-Baseline/Baseline)	Relative % Variation (Δ 12m-Baseline/Baseline)	Relative % Variation (Δ 24m-Baseline/Baseline)
Weight (kg)	110.9 ± 13.0	100.9 ± 12.3	79.1 ± 14.2	81.9 ± 14.0	−9.1 ± 2.1 ***	−29.4 ± 8.4 ***	−27.5 ± 9.6 ***
BMI (kg/m^2^)	40.7 ± 4.2	37.1 ± 4.2	28.9 ± 4.3	29.8 ± 4.5	−9.1 ± 2.1 ***	−29.4 ± 8.4 ***	−27.6 ± 9.8 ***
Calcium homeostasis							
Calcium (mmol/L)	2.37 ± 0.08	2.38 ± 0.08	2.34 ± 0.08	2.34 ± 0.08	0.5 ± 3.5	−0.9 ± 3.0 **	−1.4 ± 3.2 ***
Phosphorus (mmol/L)	1.03 ± 0.16	1.02 ± 0.15	1.12 ± 0.15	1.05 ± 0.16	0.5 ± 15.2	11.3 ± 18.7 ***	3.3 ± 18.5
iPTH (pg/mL)	48.2 ± 20.0	44.0 ± 16.3	43.7 ± 13.1	47.7 ± 16.2	−3.8 ± 31.2 *	−3.1 ± 31.2	9.1 ± 48.2
25(OH)D3 (ng/mL)	24.1 ± 10.1	24.3 ± 9.6	31.3 ± 11.4	29.7 ± 11.1	3.3 ± 19.0	34.4 ± 39.8 ***	34.0 ± 57.9 ***
Hypovitaminosis D n (%)							
Deficiency	33 (39.8)	30 (37.5)	11 (14.9)	11 (18.3)			
Insufficiency	30 (36.1)	30 (37.5)	24 (32.4)	23 (38.3)			
Normal	20 (24.1)	20 (25.0)	39 (52.7)	26 (43.3)			
Albumin (g/L)	45.0 ± 4.6	44.1 ± 2.5	44.2 ± 2.8	45.1 ± 4.5	−1.1 ± 7.2	−0.8 ± 7.2	1.2 ± 8.5
Bone markers							
Osteocalcin (ng/mL)	19.0 ± 7.5	20.2 ± 7.9	27.2 ± 8.4	22.8 ± 8.1	10.3 ± 23.8 ***	59.8 ± 61.5 ***	37.2 ± 56.8 ***
CTX (ng/mL)	0.25 ± 0.19	0.43 ± 0.20	0.51 ± 0.23	0.35 ± 0.19	112.8 ± 100.1 ***	194.4 ± 223.5 ***	108.8 ± 182.4 ***
Periostin (pmol/L)	577.1 ± 143.7	632.7 ± 153.9	554.7 ± 144.3	556.5 ± 136.0	12.0 ± 21.0 ***	−0.7 ± 20.7	−1.0 ± 22.7
Sclerostin (pmol/L)	103.3 ± 55.3	111.9 ± 64.6	98.6 ± 50.9	96.5 ± 47.7	12.1 ± 24.6 ***	−1.5 ± 19.4	−11.2 ± 17.1 ***
Semaphorin (pmol/L)	315.0 ± 91.6	307.5 ± 68.1	260.9 ± 47.1	251.8 ± 42.9	2.6 ± 18.9	−10.5 ± 16.2 ***	−11.8 ± 28.8 ***

Legend. Data are presented as mean ± SD (standard deviation). BMI: body mass index; iPTH: intact parathyroid hormone; 25(OH)D3: 25 vitamin D; CTX: type I-C telopeptide breakdown products. Measure Δ 1m-baseline/baseline represents the relative % difference between values at 1 month and baseline. Measure Δ 12m-baseline/baseline represents relative % difference between values at 12 months and baseline. Measure Δ 24m-baseline/baseline represents relative % difference between values at 24 months and baseline Relative % variation was defined as [100 × (measure 2 − measure 1)/measure 1]; * indicates a significant variation for *p* < 0.05, ** for *p* < 0.01, and *** for *p* < 0.001.

**Table 2 nutrients-15-04310-t002:** Areal bone mineral density of the patients at baseline, 12 and 24 months after sleeve gastrectomy.

	Baseline (n = 83)	12 Months (n = 76)	24 Months (n = 60)	Relative % Variation (Δ 12m-Baseline/Baseline)	Relative % Variation (Δ 24m-Baseline/Baseline)
Whole body					
aBMD (g/cm^2^)	1.187 ± 0.098	1.225 ± 0.110	1.176 ± 0.113	2.4 ± 2.9 ***	−2.4 ± 5.4 **
T-score (SD)	0.71 ± 1.02	1.14 ± 1.02	0.54 ± 1.24		
Z-score (SD)	0.63 ± 0.87	0.99 ± 0.92	0.55 ± 1.05		
Total hip					
aBMD (g/cm^2^)	1.083 ± 0.133	1.020 ± 0.137	1.017 ± 0.133	−6.6 ± 4.2 ***	−7.9 ± 4.6 ***
T-score (SD)	0.84 ± 0.97	0.43 ± 0.93	0.36 ± 0.90		
Z-score (SD)	1.09 ± 0.97	0.66 ± 0.96	0.69 ± 0.91		
Osteoporosis n (%)	0 (0%)	0 (0%)	0 (0%)		
Osteopenia n (%)	2 (2.4%)	5 (6.6%)	5 (8.3%)		
Femoral neck					
aBMD (g/cm^2^)	0.920 ± 0.150	0.874 ± 0.141	0.871 ± 0.140	−5.4 ± 7.0 ***	−5.8 ± 7.6 ***
Lumbar spine					
aBMD (g/cm^2^)	1.087 ± 0.139	1.095 ± 0.142	1.107 ± 0.159	−0.1 ± 4.1	−0.4 ± 5.0
T-score (SD)	0.58 ± 1.31	0.63 ± 1.30	0.71 ± 1.44		
Z-score (SD)	0.86 ± 1.27	0.92 ± 1.31	1.11 ± 1.4		
Osteoporosis n (%)	1 (1.2%)	0 (0%)	1 (1.7%)		
Osteopenia n (%)	9 (10.8%)	8 (10.5%)	8 (13.3%)		
Radius					
aBMD (g/cm^2^)	0.641 ± 0.061	0.638 ± 0.061	0.632 ± 0.069	−0.8 ± 1.6 **	−2.1 ± 2.4 ***
T-score (SD)	1.71 ± 1.36	1.67 ± 1.25	1.43 ± 1.41		
Z-score (SD)	2.11 ± 1.45	2.08 ± 1.40	1.95 ± 1.54		

Legend. Data are presented as mean ± SD (standard deviation). aBMD: areal bone mineral density. The T-score describes the number of SDs by which aBMD differs from the mean value expected in young healthy individuals of the same sex, whereas the Z score describes the number of SDs by which aBMD differs from the mean value expected for age and sex. Measure Δ 12m-baseline/baseline represents relative % difference between values at 12 months and baseline. Measure Δ 24m-baseline/baseline represents relative % difference between values at 24 months and baseline. Relative % variation was defined as [100 × (measure 2 − measure 1)/measure 1]; ** indicate a significant variation for *p* < 0.01, and *** for *p* < 0.001.

## Data Availability

The data used in the present analysis can be obtained through request to the corresponding author.

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
