# Peer review of "Effect of Nutritional Deprivation after Sleeve Gastrectomy on Bone Mass, Periostin, Sclerostin and Semaphorin 4D: A Two-Year Longitudinal Study"

_nutrients, 2023, doi:10.3390/nu15204310_

Round 1

Reviewer 1 Report

Your manuscript focuses on (areal) bone mineral density (aBMD) and serum/plasma markers of obese patients after bariatric surgery. The study is of high interest for the scientific field due to unknown factors that can lead to reduction of (a)BMD. Attached you will find some major/minor issues regarding your manuscript.

Major/Minor Issues:

1.     “Obesity Reference Centre”. Please indicate where the mentioned patients were treated, and which ethic committee was responsible.

2.     Please indicate significances in Figure 1 and Figure 2.

Good English language.

Author Response

Reviewer 1

Your manuscript focuses on (areal) bone mineral density (aBMD) and serum/plasma markers of obese patients after bariatric surgery. The study is of high interest for the scientific field due to unknown factors that can lead to reduction of (a)BMD. Attached you will find some major/minor issues regarding your manuscript.

Thank you very much for your very complimentary comments. They are like a crown to a massive work that brought together many teams.

Major/Minor Issues:

  1. “Obesity Reference Centre”. Please indicate where the mentioned patients were treated, and which ethic committee was responsible.

Thank you for this pertinent comment: As suggested, we have mentioned in the revised manuscript where the patients were treated: “Obesity Reference Centre, CHU Montpellier, France”. The ethic committee is now also specified: “Sud Méditerranée I, Marseille, France”.

  1. Please indicate significances in Figure 1 and Figure 2.

Thank you for the comment. As recommended, the significance (p-values) has been added in Figure 1 (Figure 2 in the revised manuscript) and in Figure 2 (Figure 3 in the revised manuscript).

Reviewer 2 Report

The paper titled: Effect of nutritional deprivation after sleeve gastrectomy on bone mass, periostin, sclerostin and semaphorin 4D: A two-year longitudinal study prepared by Laurent MAimoun et al.  falls within the scope of Nutrients Journal. It is a  quite interesting paper. The main findings showed a persistent decrease in areal bone mineral density ( aBMD) throughout the 24 months after sleeve gastrectomy (SG), but with a site-dependent effect. The bone loss was concomitant to an uncoupling between bone formation and bone resorption activities that persisted despite the attainment of a new steady state in weight. Periostin, sclerostin, and SEMA4D levels were also modified after SG, but with different time lags.

It should be underlined that this is one of the first studies that estimated this problem. The Authors have presented sufficient data. The appropriate tables and figures have been provided. The article is easy to read and logically structured.  The Authors used appropriate statistical methods. The methods are adequately described. The conclusions are consistent with the presented evidence and arguments. The Authors also added a very good section- limitations and strengths. Another strength of this manuscript is the presentation of the present study's clinical implications. The manuscript is appropriately referenced. References are up-to-date and complete. In my opinion, the presented paper may be published in the current version

Author Response

Reviewer 2

The paper titled: Effect of nutritional deprivation after sleeve gastrectomy on bone mass, periostin, sclerostin and semaphorin 4D: A two-year longitudinal study prepared by Laurent MAimoun et al.  falls within the scope of Nutrients Journal. It is a  quite interesting paper. The main findings showed a persistent decrease in areal bone mineral density ( aBMD) throughout the 24 months after sleeve gastrectomy (SG), but with a site-dependent effect. The bone loss was concomitant to an uncoupling between bone formation and bone resorption activities that persisted despite the attainment of a new steady state in weight. Periostin, sclerostin, and SEMA4D levels were also modified after SG, but with different time lags.

It should be underlined that this is one of the first studies that estimated this problem. The Authors have presented sufficient data. The appropriate tables and figures have been provided. The article is easy to read and logically structured.  The Authors used appropriate statistical methods. The methods are adequately described. The conclusions are consistent with the presented evidence and arguments. The Authors also added a very good section- limitations and strengths. Another strength of this manuscript is the presentation of the present study's clinical implications. The manuscript is appropriately referenced. References are up-to-date and complete. In my opinion, the presented paper may be published in the current version.

Thank you very much for your very complimentary comments. They are like a crown to a massive work that brought together many teams.

Reviewer 3 Report

I would like to congratulate the authors for developing the present study.

I have some questions:

1) Insert a flowchart based on CONSORT

2) Follow the CONSORT in all sections

3) Insert the practical applications / clinical applications / epidemiological applications of the present study

4) Insert information about the drop-out rate

5) Discuss aspects linked to health promotion intervention, changes in lifestyle, healthy nutrition, physical activity, and mental health to maintain aspects related to weight loss after bariatric surgery. 

Author Response

Reviewer 3

I would like to congratulate the authors for developing the present study.

Thank you for this very positive fed-back

I have some questions:

1) Insert a flowchart based on CONSORT

Thank you for this pertinent comment that clarified the design of the study for the reader. The new figure 1 was included in the revised manuscript page 9. We hope that this new figure will meet with your approval.

Figure 1. Flowchart of the population studied.

2) Follow the CONSORT in all sections

Thank you for these comments. However, we think that the CONSORT (Consolidated Standards Of Reporting Trials) is adapted of when preparing for publication in medical journals with regard to methods, results and discussion of randomised controlled trials (RCT). (please see https://www.goodreports.org/reporting-checklists/consort/.). We think that STROBE (The Strengthening the Reporting of Observational Studies in Epidemiology) seems more adapted at our longitudinal observational study. (please see https//www.equator-networks.org/reporting-guidelines/strobe). As recommended with have included all the minor modifications requested by the STROBE. All the modifications were underlined in the revised manuscript. Additionally, we send the STROBE document that indicated where each modifications were performed (please see at the end of the reponse)

3) Insert the practical applications / clinical applications / epidemiological applications of the present study

Thank you for the comment. Accordingly, with this remarks we have added the following sentence page 15: Moreover, the persistence of a high degree of bone turnover at the end of the follow-up may also suggest that aBMD loss is likely to continue for a longer time after BS and that aBMD monitoring by DXA should thus be performed beyond 24 months in these patients

4) Insert information about the drop-out rate

Thank you for this comments. Accordingly the following sentence had been introduced in the revised manuscript in the results section page 5 “During the follow-up some patients withdrew from the study (Figure 1.) : at 1 month two patients (one use of protein supplementation before the surgery and one withdrawal of informated consent), at 12 months seven patients (one patients die, one pregnant patient and five lost to follow-up or no longer wishing to participate in the study), at 24 months sixteen patients (patients lost to follow-up or no longer wishing to participate in the study).”

5) Discuss aspects linked to health promotion intervention, changes in lifestyle, healthy nutrition, physical activity, and mental health to maintain aspects related to weight loss after bariatric surgery.

Thank you for this comment. We are aware that in our original manuscript we didn’t discuss on the potential approach that may reduce the bone loss after bariatric. This omission can leave clinicians without any perspective on how to manage their patients. After a thorough review of the literature, we realized that few studies have addressed this issue. Among potential approach, physical exercise associated or not with supplementation (Ca, vitamin D and protein) appeared as the most viable strategies to mitigate bone loss following bariatric surgery. The following sentence had been introduced in the revised manuscript: “Taking into account the potential deleterious effects of weight loss, that reduce mechanical loading applied on the skeleton, it will be interestingly to evaluate strategies restoring bone loading. Then, previous clinical trials had demonstrated that physical exercise, combined or not with adequate supplementation (vitamin D, calcium and protein) is a practical approach to reduce bone loss caused by bariatric surgery [58,59]. Other studies are necessary to demonstrate whether this approach should be generalized for all patients undergoing bariatric surgery. It remains also to determine the optimal components (frequency, intensity, type and duration) of physical exercise for improving these effects on bone health, knowing that low-dose exercise program had no favorable impact [60]”. We hope that these modifications will meet with your approval.

STROBE Statement—checklist of items that should be included in reports of observational studies

Item No

Recommendation

Page
No

Title and abstract

1

(a) Indicate the study’s design with a commonly used term in the title or the abstract

1

(b) Provide in the abstract an informative and balanced summary of what was done and what was found

1 modification

Introduction

Background/rationale

2

Explain the scientific background and rationale for the investigation being reported

2-3

Objectives

3

State specific objectives, including any prespecified hypotheses

3

Methods

Study design

4

Present key elements of study design early in the paper

3 modification

Setting

5

Describe the setting, locations, and relevant dates, including periods of recruitment, exposure, follow-up, and data collection

3 modifications

Participants

6

(a) Cohort study—Give the eligibility criteria, and the sources and methods of selection of participants. Describe methods of follow-up

Case-control study—Give the eligibility criteria, and the sources and methods of case ascertainment and control selection. Give the rationale for the choice of cases and controls

Cross-sectional study—Give the eligibility criteria, and the sources and methods of selection of participants

3 modifications

(b) Cohort study—For matched studies, give matching criteria and number of exposed and unexposed

Case-control study—For matched studies, give matching criteria and the number of controls per case

Variables

7

Clearly define all outcomes, exposures, predictors, potential confounders, and effect modifiers. Give diagnostic criteria, if applicable

3-5

Data sources/ measurement

8*

 For each variable of interest, give sources of data and details of methods of assessment (measurement). Describe comparability of assessment methods if there is more than one group

4-5

Bias

9

Describe any efforts to address potential sources of bias

4

Study size

10

Explain how the study size was arrived at

Quantitative variables

11

Explain how quantitative variables were handled in the analyses. If applicable, describe which groupings were chosen and why

Statistical methods

12

(a) Describe all statistical methods, including those used to control for confounding

5

(b) Describe any methods used to examine subgroups and interactions

(c) Explain how missing data were addressed

(d) Cohort study—If applicable, explain how loss to follow-up was addressed

Case-control study—If applicable, explain how matching of cases and controls was addressed

Cross-sectional study—If applicable, describe analytical methods taking account of sampling strategy

(e) Describe any sensitivity analyses

Continued on next page

Results

Participants

13*

(a) Report numbers of individuals at each stage of study—eg numbers potentially eligible, examined for eligibility, confirmed eligible, included in the study, completing follow-up, and analysed

5 modifications

(b) Give reasons for non-participation at each stage

5 modifications

(c) Consider use of a flow diagram

new figure 1 included

Descriptive data

14*

(a) Give characteristics of study participants (eg demographic, clinical, social) and information on exposures and potential confounders

5

(b) Indicate number of participants with missing data for each variable of interest

5

(c) Cohort study—Summarise follow-up time (eg, average and total amount)

5

Outcome data

15*

Cohort study—Report numbers of outcome events or summary measures over time

5

Case-control study—Report numbers in each exposure category, or summary measures of exposure

Cross-sectional study—Report numbers of outcome events or summary measures

Main results

16

(a) Give unadjusted estimates and, if applicable, confounder-adjusted estimates and their precision (eg, 95% confidence interval). Make clear which confounders were adjusted for and why they were included

(b) Report category boundaries when continuous variables were categorized

5-15

(c) If relevant, consider translating estimates of relative risk into absolute risk for a meaningful time period

Other analyses

17

Report other analyses done—eg analyses of subgroups and interactions, and sensitivity analyses

Discussion

Key results

18

Summarise key results with reference to study objectives

15

Limitations

19

Discuss limitations of the study, taking into account sources of potential bias or imprecision. Discuss both direction and magnitude of any potential bias

19

Interpretation

20

Give a cautious overall interpretation of results considering objectives, limitations, multiplicity of analyses, results from similar studies, and other relevant evidence

15-19

Generalisability

21

Discuss the generalisability (external validity) of the study results

15-19

Other information

Funding

22

Give the source of funding and the role of the funders for the present study and, if applicable, for the original study on which the present article is based

20

*Give information separately for cases and controls in case-control studies and, if applicable, for exposed and unexposed groups in cohort and cross-sectional studies.

Note: An Explanation and Elaboration article discusses each checklist item and gives methodological background and published examples of transparent reporting. The STROBE checklist is best used in conjunction with this article (freely available on the Web sites of PLoS Medicine at http://www.plosmedicine.org/, Annals of Internal Medicine at http://www.annals.org/, and Epidemiology at http://www.epidem.com/). Information on the STROBE Initiative is available at www.strobe-statement.org.

Reviewer 4 Report

The paper "Effect of nutritional deprivation after sleeve gastrectomy on bone mass, periostin, sclerostin and semaphorin 4D: A two-year longitudinal study" investigates the changes in areal bone mineral density (aBMD) and bone turnover markers following sleeve gastrectomy (SG) and (ii) determine the parameters associated with the aBMD variations.

I believe that the manuscript is well written. The framework clearly describe the scientific evidence that supports the hypothesis that the authors have raised. The discussion clearly describe the scientific evidence that supports their findings. The methodological approach is appropriate. 

The topic results original, and the study has the potentiality of being shared with the scientific community in the present form.

Kind regards,

Author Response

Reviewer 4

The paper "Effect of nutritional deprivation after sleeve gastrectomy on bone mass, periostin, sclerostin and semaphorin 4D: A two-year longitudinal study" investigates the changes in areal bone mineral density (aBMD) and bone turnover markers following sleeve gastrectomy (SG) and (ii) determine the parameters associated with the aBMD variations.

I believe that the manuscript is well written. The framework clearly describe the scientific evidence that supports the hypothesis that the authors have raised. The discussion clearly describe the scientific evidence that supports their findings. The methodological approach is appropriate. 

The topic results original, and the study has the potentiality of being shared with the scientific community in the present form.

Thank you for this very positive fed-back
